# ApolloConv: Multi-Scale Frequency-Aware Convolutions for Robust Multivariate Time Series Forecasting

## Abstract

Time series forecasting requires models that balance expressive power with computational efficiency. While convolutional neural networks offer efficient temporal modeling, their inherent translation invariance often misaligns with the recency bias and non-stationary dynamics present in real-world time series. We propose **ApolloConv**, a convolutional architecture that enhances temporal inductive bias through integrated time–frequency modeling. ApolloConv incorporates (i) a *multi-scale embedding stem* that captures local-to-global patterns while emphasizing recent context, (ii) a *lightweight spectral gating mechanism* that modulates periodic components in the frequency domain while preserving phase coherence, and (iii) an *adaptive dilated convolution block* that prioritizes recent time steps through logarithmically scaled receptive fields. Together, these components enable effective handling of multi-scale seasonality, trend structures, and cross-variable dependencies with near-linear complexity. Extensive experiments on benchmark datasets demonstrate that ApolloConv consistently outperforms state-of-the-art CNN-based models such as TimesNet, TVNet, and ModernTCN across both short- and long-term forecasting settings, while matching or exceeding Transformer-based counterparts with significantly lower computational cost. ApolloConv provides a robust and efficient convolutional alternative for practical time series forecasting.

## 1 Introduction

Time series forecasting is crucial in many domains (Bi et al., 2023; Wu et al., 2018; Zhang et al., 2014), where capturing temporal dependencies, multi-scale patterns, and frequency-domain characteristics is essential. While Multi-Layer Perceptron (**MLPs**). Recurrent Neural Networks (**RNNS**), **Transformers** and State Space Models (**SSMs**) have recently achieved strong performance in modeling long-range dependencies and global interactions (Wang et al., 2025b; Liu et al., 2024; Zhang & Yan, 2023; Nie et al., 2022; Zhou et al., 2022; Wu et al., 2021; Zhou et al., 2021; Li et al., 2019; Vaswani et al., 2017; Si et al., 2025; Wang et al., 2024; Li et al., 2023; Challu et al., 2023; Xu et al., 2023; Wang et al., 2025c; Gu & Dao, 2023; Lin et al., 2023), each approach has limitations: Transformers incur high computational cost for long sequences; MLPs lack explicit temporal inductive biases; RNNs face sequential computation bottlenecks and gradient instability; and SSMs, while efficient, may oversimplify non-stationary dynamics and fail to preserve frequency-domain properties such as phase coherence.

**CNNs** provide a compelling alternative due to their controllable receptive field, parameter sharing, and nearly linear computational complexity (Li et al., 2025; Luo & Wang, 2024; Wang et al., 2023; Wu et al., 2022; Liu et al., 2022a). They efficiently capture local temporal dependencies and support large-scale sequence processing. Advances such as large convolutional kernels,

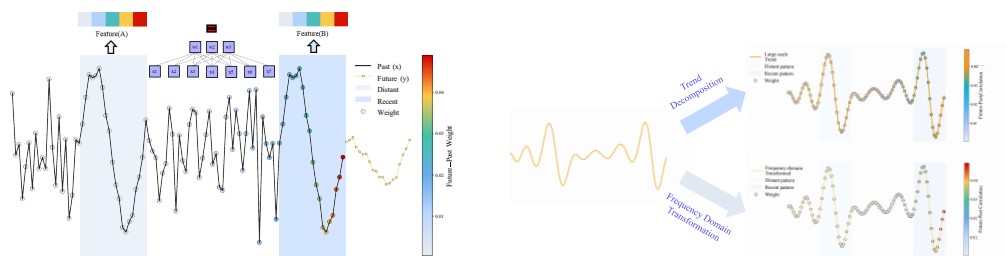

(a) CNNs leverage efficiency for time series forecasting but suffer from translation invariance, mapping similar past patterns to identical features despite varying future correlations.

(b) Trend decomposition and frequency transformation retain temporal structure, differentiating recent from distant patterns to enhance future prediction accuracy.

Figure 1: Illustration of CNN limitations and proposed solutions in time series forecasting.

dynamic weights, and patch-based embeddings further reinforce CNNs as powerful general-purpose backbones(Li et al., 2025; Luo & Wang, 2024; Wang et al., 2023; Wu et al., 2022)(Liu et al., 2022a; Lou & Yu, 2025; Wang et al., 2025a; Woo et al., 2023; Ding et al., 2022; Liu et al., 2022b; Dosovitskiy et al., 2020; Liu et al., 2021). However, the inductive bias(Battaglia et al., 2018) inherent in CNNs may not align well with time series forecasting. As illustrated in Fig. 1a, CNNs produce similar representations for two past segments with identical patterns, overlooking the fact that more recent segments usually exert a stronger influence on future outcomes. This misalignment can introduce noise and degrade forecasting accuracy, since CNNs fail to emphasize the recency of patterns that are most predictive of future trends.

To address this issue, we draw inspiration from classical time series analysis, where enhancing predictability often relies on extracting long-term trends(**?**) or applying frequency-domain transformations(Cai et al., 2024; Ye et al., 2024; Yi et al., 2023a;b; Xu et al., 2023; Zhou et al., 2022). In modern deep learning frameworks, trend extraction can be naturally realized by large-scale convolutions, while frequency-domain neural networks are widely used for capturing oscillatory dynamics. As illustrated in Fig. 1b, combining trend decomposition with frequency-domain transformations not only differentiates recent (green) and distant (beige) segments with similar past patterns but also preserves the global temporal structure and suppresses noise, ultimately improving forecasting accuracy.

Building on these observations, we propose ApolloConv, a CNN architecture for time series forecasting that refines vision-oriented convolutional biases with explicit time–frequency awareness while preserving computational efficiency. The design of ApolloConv mirrors the sequential logic of classical preprocessing yet is realized in an end-to-end deep learning framework. First, a **multi-scale convolutional stem** extracts patterns from local shocks to seasonal trends while emphasizing recency, analogous to trend extraction in traditional analysis. Second, a **frequency gating module** operates in the frequency domain, modulating magnitudes while preserving phase to capture periodicity without undermining sequential causality. Third, an adaptive dilated backbone with group-wise mixing models long-range dependencies and cross-variable interactions in a lightweight manner, prioritizing recent dynamics through logarithmically scaled dilations. Finally, a downsampling head with a second frequency gate refines temporal–spectral representations for stable long-horizon prediction. Together, these components allow ApolloConv to overcome the recency and nonstationarity limitations of conventional CNNs, achieving state-of-the-art accuracy with lower computational overhead than ModernTCN and TVNet.

**Our contributions are threefold:**

- **Trend–frequency aware CNN architecture:** We propose *ApolloConv*, which integrates multi-scale convolutions for trend extraction with frequency-domain gating for periodicity

modeling, alleviating the mismatch between CNNs' invariance bias and the requirements of time-series forecasting.

- **Lightweight temporal modeling:** Through adaptive dilated convolutions and group-wise mixing, ApolloConv exhibits low computational complexity, ensuring scalability to long sequences.

- **Forecasting Accuracy:** ApolloConv delivers transformer-level forecasting accuracy at a fraction of the computational cost, consistently outperforming other CNN-based approaches.

## 2 RELATED WORK

### 2.1 CONVOLUTIONAL ARCHITECTURES FOR TIME SERIES FORECASTING

Temporal Convolutional Networks (TCNs)(Franceschi et al., 2019; Sen et al., 2019; Bai et al., 2018) have positioned convolutional architectures as a core methodology in time series forecasting. Subsequent research has largely progressed along two main avenues: expanding the receptive field to capture long-range dependencies, and reorganizing representations to better capture temporal structures. To enlarge the receptive field, (Wang et al., 2023) introduced a multi-scale convolutional framework with cross-layer fusion, combining local and global information across different resolutions. (Liu et al., 2022a) designed a recursive downsampling-and-upsampling architecture with interactive convolutions to progressively expand the effective context. In contrast, ModernTCN employs large-kernel convolutions to directly capture extended historical patterns. Another line of work reorganizes the input representation to induce useful inductive biases. (Wu et al., 2022) reshapes 1D time series into 2D temporal patches via Fourier transforms and applies 2D convolutions to model periodicities and local variations. (Li et al., 2025) segments sequences into patches and applies dynamic 2D convolution to capture intra-patch, inter-patch, and cross-variable interactions simultaneously. Despite these advances, many convolutional designs overlook two inherent characteristics of time series: (1) the *recency bias*—where recent observations tend to have stronger predictive influence—and (2) the need for explicit handling of *nonstationarity* and *multi-periodicity* through frequency-aware operators.

### 2.2 MODELING CHANNEL DEPENDENCIES IN MULTIVARIATE FORECASTING

Multivariate forecasting methods(Ekambaram et al., 2023; Liu et al., 2023; Han et al., 2024) often trade off between fully-coupled mixing, which captures short-term cross-variable correlations but is prone to overfitting and spurious correlations due to nonstationarity and lead-lag misalignment, and channel-independent modeling(Nie et al., 2022; Xu et al., 2023), which improves robustness but may miss slow-moving long-range dependencies. Recent hybrid approaches adopt staged strategies, emphasizing intra-variable dynamics over short horizons while modeling cross-variable interactions over longer windows(Liu et al., 2024; Wang et al., 2025b). However, many such models rely on Transformer-based global attention, incurring quadratic complexity in sequence length. To address these issues, ApolloConv incorporates group-wise convolutions with a lightweight frequency-magnitude gating mechanism. This allows tunable cross-channel interaction, reduces spectral aliasing and energy drift, and maintains linear time and memory complexity.

## 3 METHODOLOGY

We propose *ApolloConv*, a CNN for time series forecasting that refines vision-oriented convolutional inductive biases, such as translation invariance, to better suit the directional, nonstationary nature of temporal data. As shown in Figure 4, *ApolloConv* embeds the input sequence with a multi-scale representation and spectral gate to capture recency and periodicity, applies an adaptive dilated block for long-range dependencies, and uses a downsampling module with a second spectral gate and linear head for efficient forecasting.

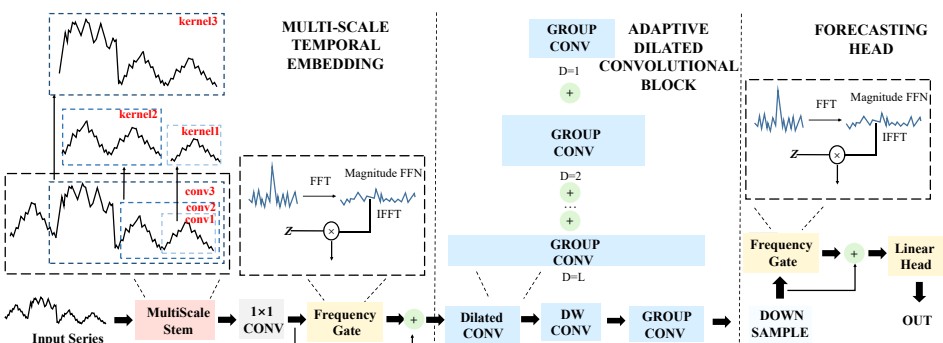

Figure 2: Model Architecture of the proposed network.

## 3.1 Multi-Scale Temporal Embedding

Standard CNNs assume translation-invariant locality, treating all temporal positions equally, which dilutes the recency bias critical for time series. To address this, we design an embedding module that captures diverse temporal resolutions while prioritizing recent dynamics, overcoming limitations of single-scale embeddings in CNNs like ModernTCN (Luo & Wang, 2024).

Given an input sequence $\mathbf{X}_{\text{in}} \in \mathbb{R}^{B \times T \times C}$ (batch size $B$, history length $T$, variables $C$), we fold variables into the batch to form $\mathbf{X}_f \in \mathbb{R}^{(BC) \times T \times 1}$. We apply $K = 3$ convolutional branches with kernel sizes $\{k_j = s \cdot 2^{2-j}\}_{j=1}^{K}$ ($k_1 = 4s, k_2 = 2s, k_3 = s$) and shared stride $s$, capturing short-to-long patterns (local shocks to seasonal trends) with emphasis on recent timesteps via small kernels. After padding $\mathbf{X}_f$ to ensure output length $N = \lceil T/s \rceil$, the multi-scale embedding is:

$$\mathbf{Z}^{(j)} = \text{LayerNorm}\Big(\text{Conv1d}_{k_j,s}\big(\text{pad}(\mathbf{X}_f)\big)\Big),$$
$$\mathbf{H} = \text{Conv1d}_{1,1}\Big(\text{concat}_{j=1}^{K}\mathbf{Z}^{(j)}\Big), \tag{1}$$

where $\mathbf{Z}^{(j)} \in \mathbb{R}^{(BC) \times U_j \times N}$, concatenation yields $\mathbb{R}^{(BC) \times (\sum U_j) \times N}$, and a point-wise convolution produces $\mathbf{H} \in \mathbb{R}^{(BC) \times D \times N}$. We reshape to $\mathbf{H}_0 \in \mathbb{R}^{B \times C \times D \times N}$. This multi-scale design counters uniform locality by weighting recent patterns more heavily, unlike ModernTCN's fixed downsampling.

To address nonstationarity and periodicities missed by time-domain CNNs like TVNet, we apply a phase-preserving spectral gate:

$$\mathbf{X} = \mathcal{F}_t(\mathbf{H}_0),$$
$$\mathbf{H}_{\text{emb}} = \mathbf{H}_0 + \boldsymbol{\gamma}_0 \odot \mathcal{F}_t^{-1}\Big(g\big(\log(1 + |\mathbf{X}|)\big) \odot e^{\mathrm{i}\angle\mathbf{X}}\Big), \tag{2}$$

where $\mathcal{F}_t(\cdot)$ and $\mathcal{F}_t^{-1}(\cdot)$ are real FFT/iFFT, $g(\cdot)$ is a group-wise 1x1 convolutional MLP (groups $= C$), $\boldsymbol{\gamma}_0 \in \mathbb{R}^{1 \times C \times 1 \times 1}$ is a learnable gate, and $\odot$ is element-wise multiplication. By modulating the magnitude spectrum while preserving phase, this gate captures global periodicities without diluting sequential causality, unlike TVNet's translation-invariant pooling.

## 3.2 Adaptive Dilated Convolutional Block

Vision-oriented CNNs apply uniform kernels, ignoring the recency bias where recent patterns outweigh distant ones. To model long-range dependencies efficiently while emphasizing recent timesteps, we design a block with adaptive dilated convolutions and group-wise mixing, tailored to sequence length and cross-variable interactions.

We reshape $\mathbf{H}_{\text{emb}} \in \mathbb{R}^{B \times C \times D \times N}$ to $\mathbb{R}^{B \times (CD) \times N}$ and compute a dilation set $\{r_u\}_{u=1}^S$, starting with $r_1 = 1$ and growing as $r_{u+1} = 2r_u$ until the receptive field reaches $T \cdot \text{rf\_ratio}$. Unlike ModernTCN's fixed large kernels, which treat all timesteps uniformly, this logarithmic scaling (S $\approx \log T$) prioritizes recent localities via small dilations. The block aggregates:

$$\mathbf{S}_{\text{agg}} = \text{LayerNorm}\Big( \phi \Big( \mathcal{D}_k \Big( \mathbf{H}_{\text{emb}} + \sum_{u=1}^S \mathcal{D}_k^{(r_u)}(\mathbf{H}_{\text{emb}}) \Big) \Big) \Big), \tag{3}$$

where $\mathcal{D}_k^{(r)}(\cdot)$ is a depthwise 1D convolution with kernel size $k$ and dilation $r$, and $\phi(\cdot)$ is GELU.

To capture time-varying cross-variable dependencies conservatively, unlike TVNet's heavy 3D mixing, we apply a lightweight group-wise feed-forward network:

$$\mathbf{X}_{\text{mix}} = \mathbf{H}_{\text{emb}} + \text{PW}_{\text{groups}=C}^{D_{\text{ff}} \to D} \Big( \phi \big( \text{PW}_{\text{groups}=C}^{D \to D_{\text{ff}}}(\mathbf{S}_{\text{agg}}) \big) \Big), \tag{4}$$

where $\text{PW}_{\text{groups}=C}^{A \to B}(\cdot)$ projects channels from $A$ to $B$ with $C$ groups, preserving per-variable dynamics while modeling lead–lag effects.

### 3.3 FORECASTING HEAD

To produce predictions efficiently, we downsample the temporal dimension and apply a second spectral gate for long-horizon stability, followed by a linear head. From $\mathbf{X}_{\text{mix}} \in \mathbb{R}^{B \times C \times D \times N}$, we reshape to $\mathbb{R}^{(BC) \times D \times N}$, apply a stride-2 convolution to reduce temporal redundancy, and reshape to $\mathbb{R}^{B \times C \times 2D \times N/2}$. A second spectral gate mitigates nonstationarity, preserving recency and periodicity:

$$\begin{aligned} \mathbf{X}_{\downarrow} &= \text{Conv1d}_{k,s=2}\big(\text{pad}(\mathbf{X}_{\text{mix}})\big), \\ \mathbf{X}_{\text{sg}} &= \mathbf{X}_{\downarrow} + \boldsymbol{\gamma}_1 \odot \mathcal{F}_t^{-1}\Big( g\big(\log(1 + |\mathcal{F}_t(\mathbf{X}_{\downarrow})|)\big) \odot e^{\mathrm{i}\angle \mathcal{F}_t(\mathbf{X}_{\downarrow})} \Big), \end{aligned} \tag{5}$$

where symbols follow Eq. equation 2. Finally, we flatten and project:

$$\hat{\mathbf{Y}} = \text{Linear}\big(\text{Flatten}_{D,t}(\mathbf{X}_{\text{sg}})\big), \tag{6}$$

where $\hat{\mathbf{Y}} \in \mathbb{R}^{B \times T_{\text{pred}} \times C}$. This lightweight head leverages rich, recency-aware features for efficient forecasting.

### 3.4 COMPLEXITY ANALYSIS.

Let $T$ be the input sequence length, $C$ the number of variables, and $D$ the embedding width. APOLLOCONV runs in near–linear time and linear space with respect to $T$. Formally, its end-to-end time complexity is $\mathcal{O}(C\,D\,T \log T)$ and the memory complexity is $\mathcal{O}(C\,D\,T)$; the $\log T$ factor comes solely from the rFFT/iFFT in the spectral magnitude gate. Throughout the paper we use $\tilde{\mathcal{O}}(\cdot)$ for ApolloConv to indicate near-linear time while ignoring the polylog factor from FFT. Compared with Transformers (typically $\mathcal{O}(T^2)$ time/space), ApolloConv has a strictly lower order. Relative to efficient/sparse attention families (Informer/Autoformer, $\mathcal{O}(T \log T)$ time and space), ApolloConv matches the near-linear time order while reducing memory to linear. Compared with convolutional baselines that incur $\mathcal{O}(T\,D^2)$ channel mixing, ApolloConv relies on depthwise/group-wise mappings and horizon-aligned dilations, avoiding quadratic coupling while maintaining accuracy (Table 1).

Table 1: Comparison of training-time and memory complexity.

| Methods | Time Complexity | Space Complexity |
|---|---|---|
| **ApolloConv (Ours)** | $\tilde{\mathcal{O}}(C\,D\,T\log T)$ | $\mathcal{O}(C\,D\,T)$ |
| ModernTCN (Luo & Wang, 2024) | $\mathcal{O}(C\,D\,k\,T\,+\,C\,D^2\,T)$ | $\mathcal{O}(C\,D\,T)$ |
| LTSF-Linear (D/NLinear) (Zeng et al., 2023) | $\mathcal{O}(C\,T^2)$ | $\mathcal{O}(T^2)$ |
| TVNet (Li et al., 2025) | $\mathcal{O}(T\,D^2)$ | $\mathcal{O}(D^2 + T\,D)$ |
| Transformer (Vaswani et al., 2017) | $\mathcal{O}(T^2)$ | $\mathcal{O}(T^2)$ |

## 4 EXPERIMENT

**Evaluation scope.** APOLLOCONV is a purely convolutional architecture specialized for time-series forecasting; we evaluate it on standard long-term and short-term forecasting benchmarks to demonstrate robustness across horizons.

**Hyperparameters.** Model performance is sensitive to hyperparameter choices. For APOLLOCONV, we adopt the search ranges reported by ModernTCN(Luo & Wang, 2024) and keep all other baselines within their officially recommended ranges to ensure a fair comparison.

**Baselines.** For long-term and short-term forecasting, we compare APOLLOCONV with strong and recent models from three families. Transformers: iTransformer(Liu et al., 2023), PatchTST(Nie et al., 2022), Crossformer(Zhang & Yan, 2023). MLPs: MTS-Mixer(Li et al., 2023), DLinear(Zeng et al., 2023), and RLinear(Zeng et al., 2023). CNNs: TimesNet(Wu et al., 2022), MICN(Wang et al., 2023), ModernTCN(Luo & Wang, 2024), and TVNet(Li et al., 2025). In addition, we include task-specific state-of-the-art (SOTA) methods as supplementary baselines to complete the comparison, ensuring a comprehensive evaluation and showing that APOLLOCONV remains competitive against the strongest published models.

### 4.1 LONG-TERM FORECASTING

**Datasets and setup.** We evaluate ApolloConv on nine widely used multivariate benchmarks: four ETT datasets(Zhou et al., 2021), Electricity(electricity, 2024), Exchange(Lai et al., 2018), Weather(weather, 2024), Traffic(traffic, 2024), and ILI(Illness, 2024). We follow the standard preprocessing and official train/validation/test splits used in prior work, and report Mean Squared Error (MSE) and Mean Absolute Error (MAE) (lower is better).

**Results.** Across nine diverse datasets, ApolloConv consistently achieves state-of-the-art or highly competitive performance, surpassing a range of MLP-, Transformer-, and CNN-based baselines in most forecasting horizons and closely matching the best contenders in others (Table 2).

Key observations include:

- Strong Long-Horizon Forecasting. ApolloConv exhibits particularly notable gains at longer prediction lengths (e.g., 336 and 720 points), where capturing extended temporal dependencies is critical. This suggests that its multi-scale convolutional design helps mitigate error propagation often observed in Transformer-based or linear models over extended horizons.

- Consistent Cross-Domain Performance. Improvements are observed across domains including ett, exchange rates, and weather, indicating robustness to both relatively stable and highly non-stationary time series.

- Competitiveness on Short Horizons. Even at shorter horizons (e.g., 96 and 192 steps)—where lightweight models such as DLinear and RLinear are often strong—ApolloConv remains highly competitive, frequently securing top-two rankings without compromising local pattern accuracy.

- Effectiveness of Convolutional Design. The results reinforce the viability of a purely convolutional approach for time series forecasting. By leveraging multi-scale receptive fields without relying on global attention, ApolloConv balances local precision with long-range context modeling, yielding reliable predictions under varied conditions.

In summary, these findings demonstrate that ApolloConv not only pushes the performance boundaries of convolutional forecasting models but also offers a simple, scalable, and effective alternative to more complex attention-based architectures.

Table 2: Long-term forecasting results averaged across four prediction horizons: $\{24, 36, 48, 60\}$ for ILI and $\{96, 192, 336, 720\}$ for the other datasets. Lower MSE/MAE indicates better performance.

| Models | | ApolloConv (Ours) | | TVnet (2025) | | PatchTST (2022) | | iTransformer (2023) | | Crossformer (2023) | | RLinear (2023) | | MTS-Mixer (2023) | | DLinear (2023) | | TimesNet (2022) | | MICN (2024) | | ModernTCN (2024) | |
|---|---|---|---|---|---|---|---|---|---|---|---|---|---|---|---|---|---|---|---|---|---|---|---|
| Metrics | | MSE | MAE | MSE | MAE | MSE | MAE | MSE | MAE | MSE | MAE | MSE | MAE | MSE | MAE | MSE | MAE | MSE | MAE | MSE | MAE | MSE | MAE |
| ETTm1 | 96 | 0.280 | 0.338 | 0.288 | 0.343 | 0.290 | 0.342 | 0.334 | 0.368 | 0.316 | 0.373 | 0.301 | 0.342 | 0.314 | 0.358 | 0.299 | 0.343 | 0.338 | 0.375 | 0.314 | 0.360 | 0.292 | 0.346 |
| | 192 | 0.317 | 0.360 | 0.326 | 0.367 | 0.332 | 0.369 | 0.377 | 0.391 | 0.377 | 0.411 | 0.355 | 0.363 | 0.354 | 0.386 | 0.335 | 0.365 | 0.371 | 0.387 | 0.359 | 0.387 | 0.332 | 0.368 |
| | 336 | 0.348 | 0.383 | 0.365 | 0.391 | 0.366 | 0.392 | 0.426 | 0.420 | 0.431 | 0.442 | 0.370 | 0.383 | 0.384 | 0.405 | 0.369 | 0.386 | 0.410 | 0.411 | 0.398 | 0.413 | 0.365 | 0.391 |
| | 720 | 0.408 | 0.412 | 0.412 | 0.413 | 0.416 | 0.420 | 0.491 | 0.459 | 0.600 | 0.547 | 0.425 | 0.414 | 0.427 | 0.432 | 0.425 | 0.421 | 0.478 | 0.450 | 0.459 | 0.464 | 0.416 | 0.417 |
| | Avg | 0.339 | 0.373 | 0.348 | 0.379 | 0.351 | 0.381 | 0.407 | 0.410 | 0.431 | 0.443 | 0.358 | 0.376 | 0.370 | 0.395 | 0.357 | 0.379 | 0.400 | 0.450 | 0.383 | 0.406 | 0.351 | 0.381 |
| ETTm2 | 96 | 0.160 | 0.250 | 0.161 | 0.254 | 0.165 | 0.255 | 0.180 | 0.264 | 0.421 | 0.461 | 0.164 | 0.253 | 0.177 | 0.259 | 0.167 | 0.260 | 0.187 | 0.267 | 0.178 | 0.273 | 0.166 | 0.256 |
| | 192 | 0.213 | 0.290 | 0.220 | 0.293 | 0.220 | 0.292 | 0.250 | 0.309 | 0.503 | 0.519 | 0.219 | 0.290 | 0.241 | 0.303 | 0.224 | 0.303 | 0.249 | 0.309 | 0.245 | 0.316 | 0.222 | 0.293 |
| | 336 | 0.268 | 0.325 | 0.272 | 0.316 | 0.274 | 0.329 | 0.311 | 0.348 | 0.611 | 0.580 | 0.273 | 0.326 | 0.297 | 0.338 | 0.281 | 0.342 | 0.312 | 0.351 | 0.295 | 0.350 | 0.272 | 0.324 |
| | 720 | 0.345 | 0.378 | 0.349 | 0.379 | 0.362 | 0.385 | 0.412 | 0.407 | 0.996 | 0.750 | 0.366 | 0.385 | 0.396 | 0.398 | 0.397 | 0.421 | 0.497 | 0.403 | 0.389 | 0.406 | 0.351 | 0.381 |
| | Avg | 0.247 | 0.311 | 0.251 | 0.311 | 0.255 | 0.315 | 0.288 | 0.332 | 0.632 | 0.578 | 0.256 | 0.314 | 0.277 | 0.325 | 0.267 | 0.332 | 0.291 | 0.333 | 0.277 | 0.336 | 0.253 | 0.314 |
| ETTh1 | 96 | 0.356 | 0.390 | 0.371 | 0.408 | 0.370 | 0.399 | 0.386 | 0.405 | 0.386 | 0.429 | 0.366 | 0.391 | 0.372 | 0.395 | 0.375 | 0.393 | 0.384 | 0.402 | 0.396 | 0.427 | 0.368 | 0.394 |
| | 192 | 0.393 | 0.410 | 0.398 | 0.409 | 0.413 | 0.421 | 0.441 | 0.436 | 0.419 | 0.444 | 0.404 | 0.431 | 0.416 | 0.426 | 0.425 | 0.416 | 0.438 | 0.404 | 0.430 | 0.453 | 0.405 | 0.413 |
| | 336 | 0.377 | 0.410 | 0.401 | 0.409 | 0.422 | 0.436 | 0.487 | 0.448 | 0.440 | 0.461 | 0.420 | 0.423 | 0.455 | 0.449 | 0.439 | 0.443 | 0.491 | 0.469 | 0.474 | 0.508 | 0.391 | 0.412 |
| | 720 | 0.430 | 0.449 | 0.458 | 0.459 | 0.447 | 0.460 | 0.503 | 0.491 | 0.519 | 0.524 | 0.442 | 0.456 | 0.475 | 0.472 | 0.442 | 0.490 | 0.521 | 0.500 | 0.460 | 0.461 | 0.450 | 0.461 |
| | Avg | 0.389 | 0.415 | 0.407 | 0.421 | 0.413 | 0.431 | 0.454 | 0.447 | 0.441 | 0.465 | 0.408 | 0.425 | 0.430 | 0.436 | 0.420 | 0.436 | 0.458 | 0.444 | 0.433 | 0.462 | 0.404 | 0.420 |
| ETTh2 | 96 | 0.246 | 0.322 | 0.263 | 0.329 | 0.274 | 0.336 | 0.297 | 0.349 | 0.628 | 0.563 | 0.262 | 0.331 | 0.307 | 0.354 | 0.289 | 0.353 | 0.340 | 0.374 | 0.289 | 0.357 | 0.263 | 0.332 |
| | 192 | 0.297 | 0.360 | 0.319 | 0.372 | 0.339 | 0.379 | 0.380 | 0.400 | 0.703 | 0.624 | 0.320 | 0.374 | 0.374 | 0.399 | 0.383 | 0.418 | 0.402 | 0.414 | 0.409 | 0.438 | 0.320 | 0.374 |
| | 336 | 0.303 | 0.368 | 0.311 | 0.373 | 0.329 | 0.380 | 0.428 | 0.432 | 0.827 | 0.675 | 0.325 | 0.386 | 0.398 | 0.432 | 0.448 | 0.465 | 0.452 | 0.452 | 0.417 | 0.452 | 0.313 | 0.376 |
| | 720 | 0.375 | 0.422 | 0.401 | 0.434 | 0.379 | 0.422 | 0.427 | 0.445 | 1.181 | 0.840 | 0.372 | 0.421 | 0.463 | 0.465 | 0.605 | 0.551 | 0.462 | 0.468 | 0.426 | 0.473 | 0.392 | 0.433 |
| | Avg | 0.305 | 0.368 | 0.324 | 0.377 | 0.330 | 0.379 | 0.383 | 0.407 | 0.835 | 0.676 | 0.320 | 0.378 | 0.386 | 0.413 | 0.431 | 0.447 | 0.414 | 0.427 | 0.385 | 0.430 | 0.322 | 0.379 |
| Electricity | 96 | 0.131 | 0.228 | 0.142 | 0.223 | 0.129 | 0.222 | 0.148 | 0.240 | 0.187 | 0.283 | 0.140 | 0.235 | 0.141 | 0.243 | 0.153 | 0.237 | 0.168 | 0.272 | 0.159 | 0.267 | 0.129 | 0.226 |
| | 192 | 0.147 | 0.241 | 0.165 | 0.241 | 0.147 | 0.240 | 0.162 | 0.253 | 0.258 | 0.330 | 0.154 | 0.248 | 0.163 | 0.261 | 0.152 | 0.249 | 0.184 | 0.289 | 0.168 | 0.279 | 0.143 | 0.239 |
| | 336 | 0.161 | 0.258 | 0.164 | 0.269 | 0.163 | 0.259 | 0.178 | 0.269 | 0.323 | 0.369 | 0.171 | 0.264 | 0.177 | 0.277 | 0.169 | 0.267 | 0.198 | 0.300 | 0.196 | 0.308 | 0.161 | 0.259 |
| | 720 | 0.197 | 0.292 | 0.190 | 0.284 | 0.197 | 0.290 | 0.225 | 0.317 | 0.404 | 0.423 | 0.209 | 0.297 | 0.212 | 0.308 | 0.233 | 0.344 | 0.220 | 0.320 | 0.203 | 0.312 | 0.191 | 0.286 |
| | Avg | 0.159 | 0.255 | 0.165 | 0.254 | 0.159 | 0.253 | 0.178 | 0.270 | 0.293 | 0.351 | 0.169 | 0.261 | 0.173 | 0.272 | 0.177 | 0.274 | 0.192 | 0.295 | 0.182 | 0.292 | 0.156 | 0.253 |
| Weather | 96 | 0.141 | 0.192 | 0.147 | 0.198 | 0.149 | 0.198 | 0.174 | 0.214 | 0.153 | 0.217 | 0.175 | 0.225 | 0.156 | 0.206 | 0.152 | 0.237 | 0.172 | 0.220 | 0.161 | 0.226 | 0.149 | 0.200 |
| | 192 | 0.184 | 0.236 | 0.194 | 0.238 | 0.194 | 0.241 | 0.221 | 0.254 | 0.197 | 0.269 | 0.218 | 0.260 | 0.199 | 0.248 | 0.220 | 0.282 | 0.219 | 0.261 | 0.220 | 0.283 | 0.196 | 0.245 |
| | 336 | 0.230 | 0.276 | 0.235 | 0.277 | 0.245 | 0.282 | 0.278 | 0.296 | 0.252 | 0.311 | 0.265 | 0.294 | 0.249 | 0.291 | 0.265 | 0.319 | 0.280 | 0.306 | 0.275 | 0.328 | 0.238 | 0.277 |
| | 720 | 0.302 | 0.326 | 0.308 | 0.331 | 0.314 | 0.334 | 0.358 | 0.347 | 0.318 | 0.363 | 0.329 | 0.339 | 0.336 | 0.343 | 0.323 | 0.362 | 0.365 | 0.359 | 0.311 | 0.356 | 0.314 | 0.334 |
| | Avg | 0.214 | 0.257 | 0.221 | 0.261 | 0.226 | 0.264 | 0.258 | 0.278 | 0.230 | 0.290 | 0.247 | 0.279 | 0.235 | 0.272 | 0.240 | 0.300 | 0.259 | 0.287 | 0.242 | 0.298 | 0.224 | 0.264 |
| Traffic | 96 | 0.382 | 0.273 | 0.367 | 0.252 | 0.360 | 0.249 | 0.395 | 0.268 | 0.512 | 0.290 | 0.496 | 0.375 | 0.462 | 0.332 | 0.410 | 0.282 | 0.593 | 0.321 | 0.508 | 0.301 | 0.368 | 0.253 |
| | 192 | 0.394 | 0.276 | 0.381 | 0.262 | 0.379 | 0.256 | 0.417 | 0.276 | 0.523 | 0.297 | 0.503 | 0.377 | 0.488 | 0.354 | 0.423 | 0.287 | 0.617 | 0.336 | 0.536 | 0.315 | 0.379 | 0.261 |
| | 336 | 0.409 | 0.286 | 0.395 | 0.268 | 0.392 | 0.264 | 0.433 | 0.283 | 0.530 | 0.300 | 0.517 | 0.382 | 0.498 | 0.360 | 0.436 | 0.296 | 0.629 | 0.336 | 0.525 | 0.310 | 0.397 | 0.270 |
| | 720 | 0.448 | 0.306 | 0.442 | 0.290 | 0.432 | 0.286 | 0.467 | 0.302 | 0.573 | 0.313 | 0.555 | 0.398 | 0.529 | 0.370 | 0.466 | 0.315 | 0.640 | 0.350 | 0.571 | 0.323 | 0.440 | 0.296 |
| | Avg | 0.408 | 0.285 | 0.396 | 0.268 | 0.391 | 0.264 | 0.428 | 0.282 | 0.535 | 0.300 | 0.518 | 0.383 | 0.494 | 0.354 | 0.434 | 0.295 | 0.620 | 0.336 | 0.535 | 0.312 | 0.396 | 0.270 |
| Exchange | 96 | 0.080 | 0.195 | 0.080 | 0.195 | 0.093 | 0.214 | 0.086 | 0.206 | 0.186 | 0.346 | 0.083 | 0.301 | 0.083 | 0.201 | 0.081 | 0.203 | 0.107 | 0.234 | 0.102 | 0.235 | 0.080 | 0.196 |
| | 192 | 0.167 | 0.289 | 0.163 | 0.285 | 0.192 | 0.312 | 0.177 | 0.299 | 0.467 | 0.522 | 0.170 | 0.293 | 0.174 | 0.296 | 0.157 | 0.293 | 0.226 | 0.344 | 0.172 | 0.316 | 0.166 | 0.288 |
| | 336 | 0.305 | 0.397 | 0.291 | 0.394 | 0.350 | 0.432 | 0.331 | 0.417 | 0.783 | 0.721 | 0.309 | 0.401 | 0.336 | 0.417 | 0.305 | 0.414 | 0.367 | 0.448 | 0.272 | 0.407 | 0.307 | 0.398 |
| | 720 | 0.657 | 0.582 | 0.658 | 0.594 | 0.911 | 0.716 | 0.847 | 0.691 | 1.367 | 0.943 | 0.817 | 0.680 | 0.900 | 0.715 | 0.643 | 0.601 | 0.964 | 0.746 | 0.714 | 0.658 | 0.656 | 0.582 |
| | Avg | 0.302 | 0.366 | 0.298 | 0.367 | 0.387 | 0.419 | 0.360 | 0.403 | 0.701 | 0.633 | 0.345 | 0.394 | 0.373 | 0.407 | 0.297 | 0.378 | 0.416 | 0.443 | 0.315 | 0.404 | 0.302 | 0.366 |
| ILI | 24 | 1.292 | 0.712 | 1.324 | 0.712 | 1.319 | 0.754 | 2.207 | 1.032 | 3.040 | 1.186 | 4.337 | 1.507 | 1.472 | 0.798 | 2.215 | 1.081 | 2.317 | 0.934 | 2.684 | 1.112 | 1.347 | 0.717 |
| | 36 | 1.150 | 0.682 | 1.190 | 0.772 | 1.430 | 0.834 | 1.934 | 0.951 | 3.356 | 1.230 | 4.205 | 1.481 | 1.435 | 0.745 | 1.963 | 0.963 | 1.972 | 0.920 | 2.507 | 1.013 | 1.250 | 0.778 |
| | 48 | 1.151 | 0.704 | 1.456 | 0.782 | 1.553 | 0.815 | 2.127 | 1.004 | 3.441 | 1.223 | 4.257 | 1.484 | 1.474 | 0.822 | 2.130 | 1.024 | 2.238 | 0.940 | 2.423 | 1.012 | 1.388 | 0.781 |
| | 60 | 1.375 | 0.796 | 1.652 | 0.796 | 1.470 | 0.788 | 2.298 | 0.998 | 3.608 | 1.302 | 4.278 | 1.487 | 1.839 | 0.912 | 2.368 | 1.096 | 2.027 | 0.928 | 2.653 | 1.085 | 1.774 | 0.868 |
| | Avg | 1.242 | 0.724 | 1.406 | 0.766 | 1.443 | 0.798 | 2.141 | 0.996 | 3.361 | 1.235 | 4.269 | 1.490 | 1.555 | 0.819 | 2.169 | 1.041 | 2.139 | 0.931 | 2.567 | 1.055 | 1.440 | 0.786 |
| 1st count | | 35 | 31 | 3 | 9 | 6 | 8 | 0 | 0 | 0 | 0 | 1 | 3 | 0 | 0 | 3 | 0 | 0 | 0 | 0 | 0 | 5 | 4 |

## 4.2 SHORT-TERM FORECASTING

**Datasets and setup.** We adopt the M4 benchmark for short-term forecasting, following the original protocol of (Makridakis et al., 2020). Unless otherwise specified, the input context length is set to *twice* the forecast horizon as in (Wu et al., 2022). To assess performance, we report the Symmetric Mean Absolute Percentage Error (SMAPE), Mean Absolute Scaled Error (MASE), and the Overall Weighted Average (OWA). To strengthen the comparative study, we include strong recent models such as TimeMixer(Wang et al., 2024) and N-HiTS(Challu et al., 2023).

**Results.**

Table 3: Short-term forecasting on the M4 dataset. We report SMAPE, MASE, and OWA (lower is better).

| | Models | ApolloConv (Ours) | TVNet (2025) | PatchTST (2022) | TimeMixer (2024) | Crossformer (2023) | RLinear (2023) | MTS-Mixer (2023) | DLinear (2023) | TimesNet (2022) | MICN (2023) | ModernTCN (2024) | N-HiTS (2023) |
|---|---|---|---|---|---|---|---|---|---|---|---|---|---|
| **Yearly** | SMAPE | **13.170** | 13.217 | 13.258 | _13.206_ | 13.392 | 13.944 | 13.548 | 16.965 | 13.387 | 14.935 | 13.226 | 13.728 |
| | MASE | 2.95 | **2.899** | 2.985 | _2.916_ | 3.001 | 3.015 | 3.091 | 4.283 | 2.996 | 3.523 | 2.957 | 3.048 |
| | OWA | _0.774_ | **0.768** | 0.786 | 0.776 | 0.787 | 0.807 | 0.803 | 1.058 | 0.786 | 0.900 | 0.777 | 0.803 |
| **Quarterly** | SMAPE | _9.985_ | 9.986 | 10.197 | 9.996 | 16.317 | 10.702 | 10.128 | 12.145 | 10.100 | 11.452 | **9.971** | 10.792 |
| | MASE | **1.159** | **1.159** | 1.803 | 1.166 | 2.197 | 1.299 | 1.196 | 1.520 | 1.182 | 1.389 | 1.167 | 1.283 |
| | OWA | _0.876_ | _0.876_ | 1.803 | **0.825** | 1.542 | 0.959 | 0.896 | 1.106 | 0.890 | 1.026 | 0.878 | 0.958 |
| **Monthly** | SMAPE | **12.366** | _12.493_ | 12.641 | 12.605 | 12.924 | 13.363 | 12.717 | 13.514 | 12.670 | 13.773 | 12.556 | 14.260 |
| | MASE | **0.906** | 0.921 | 0.930 | 0.919 | 0.966 | 1.014 | 0.931 | 1.037 | 0.933 | 1.076 | _0.917_ | 1.102 |
| | OWA | **0.855** | _0.866_ | 0.876 | 0.869 | 0.902 | 0.940 | 0.879 | 0.956 | 0.878 | 0.983 | _0.866_ | 1.012 |
| **Others** | SMAPE | **4.344** | 4.764 | 4.964 | _4.564_ | 5.493 | 5.437 | 4.817 | 6.709 | 4.891 | 6.716 | 4.715 | 4.954 |
| | MASE | **2.98** | 2.986 | _2.985_ | 3.115 | 3.690 | 3.706 | 3.255 | 4.953 | 3.302 | 4.717 | 3.107 | 3.264 |
| | OWA | **0.927** | 0.969 | 1.044 | _0.982_ | 1.160 | 1.157 | 1.02 | 1.487 | 1.035 | 1.451 | 0.986 | 1.036 |
| **WA** | SMAPE | **11.578** | _11.671_ | 11.807 | 11.723 | 13.474 | 12.473 | 11.892 | 13.639 | 11.829 | 13.130 | 11.698 | 12.840 |
| | MASE | _1.541_ | **1.536** | 1.590 | 1.559 | 1.866 | 1.677 | 1.608 | 2.095 | 1.585 | 1.896 | 1.556 | 1.701 |
| | OWA | **0.83** | _0.832_ | 0.851 | 0.840 | 0.985 | 0.898 | 0.859 | 1.051 | 0.851 | 0.980 | 0.838 | 0.918 |

## 5 MODEL ANALYSIS

### 5.1 COMPUTATION COMPLEXITY

**Results.** As shown in the efficiency comparisons for the ETTm2 dataset in Fig.3 (L=720 for MSE and L=192 for MAE), AppoloConv achieves superior performance-accuracy trade-offs in terms of training time and memory footprint. Key observations include:

- **Optimal Pareto Efficiency**. AppoloConv delivers the lowest MSE and MAE while maintaining low memory usage and moderate training times, positioning it on the efficient frontier compared to models like FEDformer and TimesNet .

- **Advantage Over High-Resource Models**. AppoloConv outperforms resource-intensive baselines such as PatchTST and MICN in accuracy with significantly lower memory and comparable or faster training, highlighting the benefits of convolution-based designs for scalable forecasting.

- **Balanced Efficiency Against Linear and Transformer Models**. Compared to efficient linear models like DLinear, AppoloConv provides better accuracy at a modest increase in time and memory; against Transformers like iTransformer, it achieves similar or lower errors with faster training, demonstrating robust computational advantages without sacrificing predictive power.

These results underscore AppoloConv's computational efficiency, enabling high-accuracy long-term forecasting on resource-constrained environments while outperforming diverse baselines in overall complexity-accuracy balance.

### 5.2 ABLATION ANALYSIS

We conduct an ablation study to evaluate the contribution of key components in *ApolloConv*. The results, shown in Table 4, highlight the impact of the multi-scale temporal embedding, adaptive dilated convolutional block, and forecasting head.

**Ablation of Forecasting Head (Downsampling + Frequency Gate)**. Removing the multi-scale temporal embedding results in a performance decrease across all datasets. This highlights the critical role of multi-scale embeddings in capturing diverse temporal patterns and enabling the model

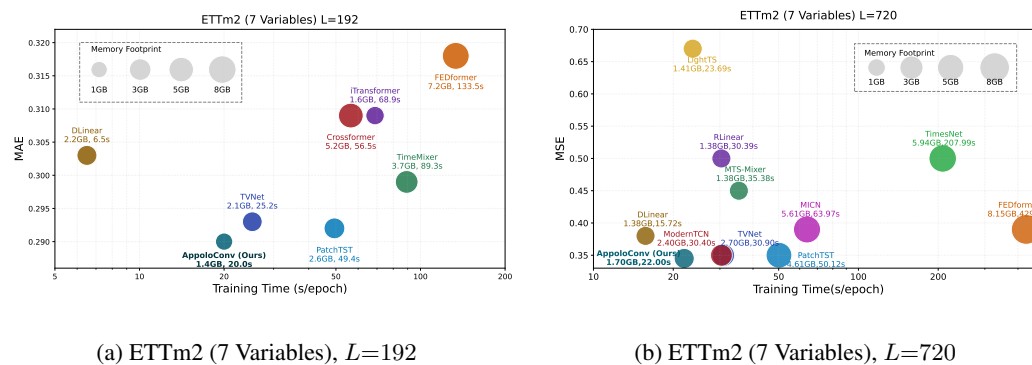

(a) ETTm2 (7 Variables), $L$=192     (b) ETTm2 (7 Variables), $L$=720

Figure 3: Model efficiency comparison on ETTm2 under the setting of L(prediction length) =192/720.

to prioritize recent dynamics over outdated patterns, which is essential for accurate time series forecasting.

**Ablation of Adaaptive Dilated Convolutional Block**. Omitting the adaptive dilated convolutional block leads to a slight performance decline, demonstrating the importance of adaptive dilations in capturing long-range dependencies while prioritizing more recent time steps. This design allows *ApolloConv* to model long-range interactions without introducing unnecessary complexity, addressing the challenges of both recency and non-stationarity in time series data.

**Ablation of Forecasting Head (Downsampling + Frequency Gate)**. Removing the forecasting head, which includes the downsampling module and the frequency gate, results in a substantial performance drop. This confirms the importance of dual denoising, where the frequency gate preserves important temporal frequencies while suppressing noise. The downsampling module further refines the temporal representations, ensuring stable long-term predictions and preventing redundancy in the temporal features.

Table 4: Ablation in **ApolloConv**.

| Datasets | ETTm1 | | ETTm2 | | ETTh1 | | ETTh2 | | Weather | | ILI | |
|---|---|---|---|---|---|---|---|---|---|---|---|---|
| | MSE | MAE | MSE | MAE | MSE | MAE | MSE | MAE | MSE | MAE | MSE | MAE |
| **ApolloConv** | **0.339** | **0.373** | **0.247** | **0.311** | **0.389** | **0.415** | **0.305** | **0.368** | **0.214** | **0.257** | **0.142** | **0.724** |
| **w/o Multi-Scale Temporal Embedding** | 0.345 | 0.376 | 0.252 | 0.315 | 0.391 | 0.417 | 0.316 | 0.373 | 0.220 | 0.265 | 1.528 | 0.815 |
| **w/o Adaptive Dilated Convolutional Block** | 0.354 | 0.376 | 0.249 | 0.312 | 0.398 | 0.421 | 0.308 | 0.369 | 0.216 | 0.257 | 1.733 | 0.889 |
| **w/o Forecasting Head** | 0.341 | 0.374 | 0.251 | 0.315 | 0.394 | 0.418 | 0.310 | 0.370 | 0.219 | 0.262 | 1.389 | 0.760 |

# 6 CONCLUSION AND FUTURE WORK

We proposed *ApolloConv*, a CNN-based model for time series forecasting that provides a solution to the limitations of traditional convolutions, such as translation invariance and noise sensitivity. By integrating multi-scale temporal embeddings and frequency-domain gating, we effectively capture recent patterns and reduce noise, while adaptive dilated convolutions model long-range dependencies efficiently. *ApolloConv* achieves superior accuracy with lower computational cost compared to existing methods. In future work, we plan to further refine this approach for more efficient, lightweight convolutional forecasting by enhancing its denoising capabilities and handling longer sequences.

ETHICS STATEMENT

This work adheres to the ICLR Code of Ethics. Our study focuses on advancing methodologies for time series forecasting using deep learning techniques, particularly aimed at improving predictive modeling for various time-dependent phenomena, such as weather patterns, financial data, and industrial sensor readings. It involves no human subjects, personally identifiable information, or sensitive user data. The data used in our experiments are publicly available time series datasets, with no direct involvement of living beings or biological data.

The proposed model aims to enhance forecasting accuracy and reliability across various domains. While the dataset and models could inform decision-making in sectors such as finance, healthcare, and energy, they do not directly enable harmful applications. Any future deployment in safety-critical domains must consider regulatory, ethical, and societal constraints beyond the scope of this work. We report all methods and results transparently and disclose no conflicts of interest or external sponsorship. All experiments were designed and conducted in accordance with standards of research integrity.

REPRODUCIBILITY STATEMENT

We guarantee the reproducibility of our results for the time series forecasting model, *ApolloConv*. All dataset construction details, including data sources, sampling strategies, preprocessing steps, and time series annotations, are provided. Task definitions, data splits, data distribution types, and evaluation metrics (MSE, MAE, SMAPE, MASE, OWA) are clearly outlined. The model architectures, hyperparameters, training schedules, and preprocessing/normalization techniques are specified, with full experimental details available in the appendix. We provide the exact time steps, sequence lengths, and data splits used, and all models are evaluated using standardized scripts. The dataset, model code, and scripts to reproduce all tables and figures will be made publicly available.

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

## A   COMPUTATIONAL COST ANALYSIS

Consider the input time series tensor $\mathbf{X} \in \mathbb{R}^{B \times C \times T}$, where $B$ is the batch size, $C$ is the number of variables, and $T$ is the sequence length. The embedding dimension of *ApolloConv* is denoted by $D$. The computational complexity for different modules can be described as follows:

CONVOLUTION LAYER

For each 1D convolution layer, with input/output embedding width $D$, kernel size $k$, and sequence length $T$:

$$\text{FLOPs}_{\text{Conv1D}} = B \times C \times T \times D \times k \tag{7}$$

$$\text{Params}_{\text{Conv1D}} = C \times D \times k \tag{8}$$

MULTI-SCALE CONVOLUTION

For the multi-scale convolution (with kernel sizes $k_1, k_2, k_3$):

$$\text{FLOPs}_{\text{MultiScale}} = B \times D \times T \times (k_1 + k_2 + k_3) \tag{9}$$

$$\text{Params}_{\text{MultiScale}} = D \times (k_1 + k_2 + k_3) \tag{10}$$

DILATED CONVOLUTION BLOCK

In the dilated convolution block, with dilation rate $r$ and kernel size $k$:

$$\text{FLOPs}_{\text{Dilated}} = B \times C \times T \times D \times k \times r \tag{11}$$

$$\text{Params}_{\text{Dilated}} = C \times D \times k \times r \tag{12}$$

GROUP FEED-FORWARD NETWORK (GROUP FFN)

For the Group Feed-Forward Network (group-wise 1x1 convolution), where $D_{\text{ff}}$ is the hidden width and the number of groups is set to $C$:

$$\text{FLOPs}_{\text{Group FFN}} = B \times C \times T \times \frac{D \times D_{\text{ff}} + D_{\text{ff}} \times D}{C} \tag{13}$$

$$\text{Params}_{\text{Group FFN}} = \frac{C \times (D \times D_{\text{ff}} + D_{\text{ff}} \times D)}{C} = 2D \times D_{\text{ff}} \tag{14}$$

FREQUENCY DOMAIN GATE (FFT-BASED)

For the frequency domain gate, we apply FFT-based operations:

$$\text{FLOPs}_{\text{FFT}} = B \times C \times T \times \log T \tag{15}$$

The parameter count is negligible (FFT/iFFT are non-learnable), so:

$$\text{Params}_{\text{FFT}} = 0 \tag{16}$$

TOTAL COMPUTATIONAL COMPLEXITY

The total FLOPs is the sum of all components:

$$\text{FLOPs}_{\text{Total}} = \text{FLOPs}_{\text{Conv1D}} + \text{FLOPs}_{\text{MultiScale}} + \text{FLOPs}_{\text{Dilated}} + \text{FLOPs}_{\text{Group FFN}} + \text{FLOPs}_{\text{FFT}} \tag{17}$$

The total parameter count is:

$$\text{Params}_{\text{Total}} = \text{Params}_{\text{Conv1D}} + \text{Params}_{\text{MultiScale}} + \text{Params}_{\text{Dilated}} + \text{Params}_{\text{Group FFN}} \tag{18}$$

# B DATASETS

## B.1 LONG-TERM FORECAST DATASETS

To evaluate the long-term forecasting capabilities, we used nine widely recognized real-world datasets, covering domains such as weather, traffic, electricity, exchange rates, influenza-like illness (ILI), and the four Electricity Transformer Temperature (ETT) datasets (ETTh1, ETTh2, ETTm1, ETTm2). For the imputation task, benchmark datasets were established using datasets from weather, electricity, and the four ETT datasets. These datasets, widely used in the field, cover various aspects of daily life.

The characteristics of each dataset, including the total number of timesteps, the count of variables, and the sampling frequency, are summarized in Table 5. The datasets are partitioned into training, validation, and testing subsets in chronological order, with the Electricity Transformer Temperature (ETT) dataset employing a 6:2:2 ratio and the remaining datasets using a 7:1:2 ratio. Normalization to a zero mean is applied to the training, validation, and testing subsets based on the mean and standard deviation of the training subset. Each dataset comprises a single, continuous, long-time series, with samples extracted using a sliding window technique.

Further details regarding the datasets are as follows:

- **Weather**[1] consists of 21 climatic variables, such as humidity and air temperature, recorded in Germany throughout 2020.

- **Traffic**[2] includes road occupancy rates collected by 862 sensors across San Francisco Bay area highways over a two-year period, provided by the California Department of Transportation.

- **Electricity**[3] comprises hourly electricity usage data for 321 consumers from 2012 to 2014.

- **Exchange**[4] encompasses daily exchange rates for eight currencies, observed from 1990 to 2016.

- **ILI**[5], which stands for Influenza-Like Illness, contains weekly counts of ILI patients in the United States from 2002 to 2021. It includes seven metrics, such as ILI patient counts across various age groups and the proportion of ILI patients relative to the total patient population. The data is provided by the Centers for Disease Control and Prevention of the United States.

- **ETT**[6] (The Electricity Transformer Temperature) dataset comprises data from seven sensors across two Chinese counties, featuring load and oil temperature metrics. It includes four subsets: 'ETTh1' and 'ETTh2' for hourly data, and 'ETTm1' and 'ETTm2' for 15-minute intervals.

Table 5: Dataset descriptions of long-term forecasting and imputation.

| Dataset | Weather | Traffic | Exchange | Electricity | ILI | ETTh1 | ETTh2 | ETTm1 | ETTm2 |
|---|---|---|---|---|---|---|---|---|---|
| Dataset Size | 52696 | 17544 | 7207 | 26304 | 966 | 17420 | 17420 | 69680 | 69680 |
| Variable Number | 21 | 862 | 8 | 321 | 7 | 7 | 7 | 7 | 7 |
| Sampling Frequency | 10 mins | 1 hour | 1 day | 1 hour | 1 week | 1 hour | 1 hour | 15 mins | 15 mins |

---

[1]https://www.bgc-jena.mpg.de/wetter/

[2]https://pems.dot.ca.gov/

[3]https://archive.ics.uci.edu/dataset/321/electricityloaddiagrams20112014

[4]https://github.com/laiguokun/multivariate-time-series-data

[5]https://github.com/laiguokun/multivariate-time-series-data

[6]https://github.com/zhouhaoyi/ETDataset

## B.2 SHORT-TERM FORECAST DATASETS

The M4 dataset, which includes 100,000 heterogeneous time series from various domains, presents a unique challenge for short-term forecasting. This dataset, drawn from diverse fields, showcases the variability in temporal patterns and distinct characteristics across different time series.

Table 6 provides a detailed overview of the M4 dataset, outlining the number of samples in both the training and test sets, the number of variables per series, and the prediction length for each subset.

Table 6: Dataset descriptions of M4 forecasting

| Dataset | Sample Numbers (train set, test set) | Variable Number | Prediction Length |
|---|---|---|---|
| M4 Yearly | (23000, 23000) | 1 | 6 |
| M4 Quarterly | (24000, 24000) | 1 | 8 |
| M4 Monthly | (48000, 48000) | 1 | 18 |
| M4 Weekly | (359, 359) | 1 | 18 |
| M4 Daily | (4227, 4227) | 1 | 48 |
| M4 Hourly | (414, 414) | 1 | 48 |

# C EXPERIMENT DETAILS

## C.1 LONG-TERM FORECASTING

**Implementation Details**. Our method is trained using the L2 loss, with the ADAM (Adam et al., 2014) optimizer and an initial learning rate of $310^{-3}$. We use mean square error (MSE) and mean absolute error (MAE) as evaluation metrics, and all experiments are repeated 5 times with different seeds. The final reported results are the means of these experiments. The model is implemented in PyTorch (Paszke et al., 1912) and conducted on NVIDIA A100 40GB GPUs.

The experimental setup follows the same parameters for prediction lengths $T \in \{24, 36, 48, 60\}$ for the ILI dataset and $T \in \{96, 192, 336, 720\}$ for other datasets, as specified in (Li et al., 2025). We collect baseline results from (Li et al., 2025), where all baseline models are re-executed with varying input lengths $L$, and the best results are chosen to avoid underestimating the baselines. For other models, we follow the official implementations and run them with varying input lengths $L \in \{36, 48, 96\}$ for ILI, and $L \in \{96, 192, 256, 336, 512, 720\}$ for other datasets.

**Model Parameters**. In *ApolloConv*, the default settings are as follows:

- The model consists of *ApolloConv* with hidden state dimension $D$ as a hyperparameter.

- The group-wise feed-forward network ratio is 1 and Groups $= C$.

- The kernel sizes are set as $k = \{2, 4, 8\}$, and the default stride is 2.

For baseline models, we adhere to the original parameters used in the papers. If the original papers perform long-term forecasting experiments, we follow their recommended configurations. We then rerun these models with varying input lengths and select the best results for comparison.

**Metric**. We adopt the mean square error (MSE) and mean absolute error (MAE) to evaluate long-term forecasting.

$$\text{MSE} = \frac{1}{T} \sum_{i=0}^{T} (\hat{x}_i - x_i)^2$$

$$\text{MAE} = \frac{1}{T} \sum_{i=0}^{T} |\hat{x}_i - x_i|$$

## C.2 SHORT-TERM FORECASTING

**Implementation Details**. Our method is trained with the SMAPE loss, using the ADAM (Adam et al., 2014) optimizer with an initial learning rate of $3 \times 10^{-3}$. The default training process is 100 epochs with proper early stopping. The symmetric mean absolute percentage error (SMAPE), mean absolute scaled error (MASE), and overall weighted average (OWA) are used as metrics. All experiments are repeated 5 times with different seeds and the means of the metrics are reported as the final results. Following (Wu et al., 2022), we fix the input length to be 2 times of prediction length for all models. Since the M4 dataset only contains univariate time series, we remove the cross-variable component in Crossformer.

**Model Parameter**. The *ApolloConv* model utilizes various hyperparameters depending on the dataset. Below are the typical settings used across different datasets:

- The hidden state dimension $D = 256$.
- The group-wise feed-forward network ratio Groups $= C$.
- The kernel sizes are set as $k = \{11\}$.
- The stride is set to 2.
- The dropout rate is $0.1$.
- The initial learning rate is set to $0.0003$.

These default settings are designed to work across various datasets, but they can be adjusted for specific use cases. The specific hyperparameters like the batch size, number of layers, and learning rate are adjusted depending on the dataset, ensuring that the model scales well across various time series forecasting tasks.

**Metric** For the short-term forecasting, following (Oreshkin et al., 2019), we adopt the symmetric mean absolute percentage error (SMAPE), mean absolute scaled error (MASE) and overall weighted average (OWA) as the metrics, which can be calculated as follows:

$$\text{SMAPE} = \frac{200}{T} \sum_{i=1}^{T} \frac{|X_i - \hat{X}_i|}{|X_i| + |\hat{X}_i|}$$

$$\text{MAPE} = \frac{100}{T} \sum_{i=1}^{T} \frac{|X_i - \hat{X}_i|}{|X_i|}$$

$$\text{MASE} = \frac{1}{T} \sum_{i=1}^{T} \frac{|X_i - \hat{X}_i|}{\frac{1}{T-p} \sum_{j=p+1}^{T} |X_j - X_{j-p}|}$$

$$\text{OWA} = \frac{1}{2} \left[ \frac{\text{SMAPE}}{\text{SMAPE}_{\text{Naive2}}} + \frac{\text{MASE}}{\text{MASE}_{\text{Naive2}}} \right]$$

where $p$ is the periodicity of the data. $\hat{X} \in \mathbb{R}^{T \times M}$ are the M variables' prediction results of length $T$ and corresponding ground truth. $X_i$ means the $i$-th time step in the prediction result.

## D SHOWCASES

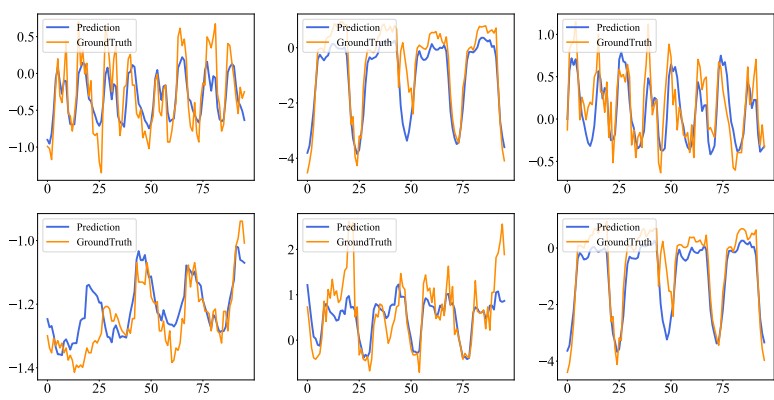

Figure 4: Visualization of ETTh1 Univariate forecasting results.

# E    LLM USE DISCLOSURE

We used large language models (LLMs) solely for assistance with grammar and wording edits, minor LaTeX formatting for tables and figures, and support in plotting. LLMs were *not* used for generating scientific claims, designing or running experiments, analyzing results, creating or altering data, or drafting substantive technical content related to time series forecasting.

All scientific content, methodologies, analyses, and conclusions were authored and independently verified by the authors. No confidential submission materials were provided to third-party LLM services. We take full responsibility for the submission and its contents.

