# OpenReview forum: "AppoloConv: Multi-Scale Frequency-Aware Convolutions for Robust Multivariate Time Series Forecasting"
_ICLR.cc/2026/Conference — ICLR 2026 Conference Withdrawn Submission_

### Official Review · Reviewer_sF7T · 2025-10-24

**Soundness:** 2
**Presentation:** 2
**Contribution:** 2
**Rating:** 2
**Confidence:** 5

**Summary:**

The paper proposes ApolloConv, a CNN-based model for robust multivariate time series forecasting. It addresses key limitations of traditional convolutional models in time series forecasting, including translation invariance and noise sensitivity, through three core components: a multi-scale embedding stem that captures local-to-global temporal patterns, a spectral gating mechanism that filters periodic noise in the frequency domain, and an adaptive dilated convolution block that models long-term dependencies efficiently. Experiments show that ApolloConv achieves state-of-the-art performance with lower complexity than prior CNN-based models.

**Strengths:**

1.	ApolloConv provides better accuracy than lightweight linear models and competitive performance to Transformer-based models, with faster training and reduced complexity.
2.	ApolloConv consistently achieves superior accuracy across both long-term and short-term forecasting benchmarks on multiple datasets.
3.	The proposed multi-scale temporal embedding effectively overcomes the fixed downsampling limitation of ModernTCN.

**Weaknesses:**

1. Formatting issues: The method name “ApolloConv” is misspelled as “AppoloConv” multiple times throughout the manuscript, including in the title, and appears in inconsistent formats (all caps, italics, or regular text) and in Section 5.2 the term “Adaptive Dilated Convolutional Block” is written as “Adaaptive.” Section 3.4 includes an unnecessary period in the title, and figure references alternate between “Fig.” and “Figure.” There is an unresolved citation (“trends(?)”) in the Introduction.
2. Similarity in experimental description: The Implementation Details section (Section C.2, “Short-Term Forecasting”) shows some overlap in phrasing with the corresponding section of the ICLR 2024 Spotlight paper ModernTCN: A Modern Pure Convolution Structure for General Time Series Analysis (Luo & Wang, 2024), likely due to similarities in experimental setup and reporting style.
3. Lack of novelty: The paper lacks strong novelty, as the proposed architecture mainly combines existing convolutional and frequency-domain techniques without introducing a fundamentally new idea.
4. Missing figure reference: Figure 2 is not referenced or described in the text. The paper should briefly explain the overall architecture and data flow corresponding to this figure.
5. Unclear notation: The notation U_jin Eq. (1) is not explicitly defined. It is unclear whether it denotes the number of output channels or the temporal length after convolution.
6. Complexity explanation issue: The complexity terms in Table 1 lack derivation or justification (e.g., why TVNet is O(TD^2)instead of O(TD)). Providing brief reasoning or references for each method’s complexity would improve clarity and fairness of comparison.
7. Issues in Ablation Study: The purpose and setup of each ablation are not clearly described—for example, it is unclear what specific components are removed in “w/o Forecasting Head” or “w/o Adaptive Dilated Block.” Table 4 lists results but provides no quantitative or causal analysis of the observed changes, and the discussion only repeats that “performance decreases.”

**Questions:**

See above

---

### Official Review · Reviewer_vVkP · 2025-10-29

**Soundness:** 3
**Presentation:** 3
**Contribution:** 3
**Rating:** 6
**Confidence:** 5

**Summary:**

This paper proposes a novel time series forecasting framework that integrates trend decomposition with frequency-domain transformations. The method introduces a Multi-Scale Temporal Embedding module to capture recent and long-term dynamics, and a phase-preserving spectral gate to enhance periodic feature modeling. Furthermore, an Adaptive Dilated Convolutional Block expands the receptive field logarithmically to balance locality and efficiency. Experiments across several benchmark datasets show consistent improvements over recent CNN- and transformer-based baselines.

**Strengths:**

1. The paper provides a clear motivation for combining time-domain trend extraction with frequency-domain modulation.
2. The model design achieves efficiency improvements over transformer-based methods by relying on lightweight convolutional and spectral operations.
3. The paper benchmarks across multiple datasets, showing consistent performance gains, which support the generalization ability of the proposed method.
4. The proposed phase-preserving spectral gate is a novel sound attempt to capture global periodicity while maintaining causal structure — a meaningful contribution compared to conventional spectral filters.
5. The combination of trend decomposition (time domain) and spectral gating (frequency domain) enables the network to capture complementary dynamics that single-domain models typically miss.

**Weaknesses:**

1. Can you provide ablations that separately remove (i) the multi-scale CNN and (ii) the phase-preserving spectral gate in Section 3.1 (Multi-Scale Temporal Embedding)? Is there experimental evidence showing that phase-preserving spectral gating outperforms standard spectral filtering? Does fixing the phase limit the model’s ability to capture frequency drift in non-stationary signals?
2. What concrete advantage does the logarithmic dilation strategy ($S \approx \log T$) bring compared to standard dilation schemes? Prior works also employ dilated convolutions — how does your approach differ? Could you include comparative experiments or replace the dilated convolution modules in these baselines to validate the effectiveness and superiority of your strategy?

3. In the motivation, you state that combining trend decomposition with frequency-domain transformations allows the model to distinguish recent (green) and distant (beige) segments with similar past patterns. Could you provide experiments or visualizations demonstrating that your module indeed differentiates between near-term and long-term similar patterns? Moreover, in the Adaptive Dilated Convolutional Block, why does logarithmic scaling (“this logarithmic scaling prioritizes recent localities via small dilations”) favor recent information, given that dilation growth expands the receptive field? Is this empirically validated?
4. In the expression “until the receptive field reaches $T \cdot$ rf_ratio,” how is this ratio determined? The manuscript does not clearly describe its selection process — is there any hyperparameter sensitivity analysis provided?
5. Why is the second spectral gate necessary in FORECASTING HEAD? What issue does it address compared with the first gate in the embedding stage?
6. In Section 5.2, Ablation Analysis, the first experiment is misnamed. It should read “Ablation of Multi-Scale Temporal Embedding”, not “Ablation of Forecasting Head (Downsampling + Frequency Gate)”.

**Questions:**

See **Weaknesses**.

---

### Official Review · Reviewer_SLAf · 2025-10-30

**Soundness:** 2
**Presentation:** 2
**Contribution:** 2
**Rating:** 2
**Confidence:** 4

**Summary:**

The paper proposes ApolloConv, a CNN-based architecture for time series forecasting that combines multi-scale convolutions for trend extraction with frequency-domain gating for periodicity modeling. The design also incorporates adaptive dilated convolutions and group-wise mixing to model long-range dependencies while emphasizing recent dynamics. The authors claim that ApolloConv addresses CNN limitations such as recency bias and nonstationarity, achieving performance comparable to Transformers but with lower computational cost. Experiments show modest improvements over other CNN-based baselines.

**Strengths:**

1. The paper highlights an intuitive limitation of CNNs (translation invariance overlooking recency) and proposes a solution that is easy to understand.
2. The architecture is engineering-wise reasonable and scalable, combining multi-scale convolutions, frequency-domain gating, and adaptive dilations in a coherent pipeline.
3. Experimental results indicate slight improvements over existing CNN-based models with lower computational cost, which may be useful in practice.

**Weaknesses:**

1. The problem statement is not clearly defined. It’s unclear whether the paper aims to solve the inefficiency of attention models, the lack of recency awareness in CNNs, or the handling of nonstationarity. The motivation feels scattered.
2. The proposed method lacks real novelty. Multi-scale convolutions and frequency-domain processing are already widely used in recent CNN-based forecasting models, so it’s hard to see what is fundamentally new here.
3. The claim that ApolloConv overcomes “recency” and “nonstationarity” limitations is not convincingly supported by experiments — there’s no specific analysis or ablation verifying these effects.
4. The performance gain over strong CNN baselines is marginal (around 1%), which makes the practical significance questionable.
5. The paper contains typos and some figures (e.g., Fig. 1) are unclear, which makes parts of the explanation hard to follow.

**Questions:**

1. Please report multi-run results (mean and std) to demonstrate the robustness of the proposed method.
2. How exactly does ApolloConv address the recency bias and nonstationarity of CNNs, and can the authors provide targeted ablation or analysis to support this claim?

---

### Note · Authors · 2025-11-24

I have read and agree with the venue's withdrawal policy on behalf of myself and my co-authors.